# Post-Ablation Stroke Despite NOAC Use: Successful Reperfusion Therapy After Dabigatran Reversal with Incidental Discovery of a Large MCA Aneurysm—A Case Report

**DOI:** 10.3390/neurolint17120190

**Published:** 2025-11-21

**Authors:** Santi Mitra Sari, Wei-Tso Chen, Chien-Hui Lee, Nai-Hsin Huang, Phyo-Wai Thu, Ling-Lun Teoh, Yu-Mei Wu, An-Bang Liu

**Affiliations:** 1Department of Neurology, Hualien Tzu Chi Hospital, No. 707, Sec. 3, Chung-Yang Rd., Hualien City 970473, Taiwan; 2Division of Cardiology, Department of Medicine, Hualien Tzu Chi Hospital, No. 707, Sec. 3, Chung-Yang Rd., Hualien City 970473, Taiwan; 3Department of Neurosurgery, Hualien Tzu Chi Hospital, No. 707, Sec. 3, Chung-Yang Rd., Hualien City 970473, Taiwan; 4Department of Medicine, School of Medicine, Tzu-Chi University, Hualien, No. 701, Sec. 3, Chung-Yang Rd., Hualien City 970374, Taiwan; 5Department of Medicine, Hualien Tzu Chi Hospital, No. 707, Sec. 3, Chung-Yang Rd., Hualien City 970473, Taiwan; 6Department of Nursing, Hualien Tzu Chi Hospital, No. 707, Sec. 3, Chung-Yang Rd., Hualien City 970473, Taiwan

**Keywords:** atrial fibrillation, catheter ablation, non–vitamin K oral anticoagulants, idarucizumab, intravenous thrombolysis, endovascular thrombectomy, aneurysm, stent embolization

## Abstract

**Background:** Catheter ablation is an established rhythm-control strategy for atrial fibrillation (AF), yet peri-procedural embolic stroke may still occur despite uninterrupted NOAC therapy. **Case presentation:** A 49-year-old woman on dabigatran developed acute ischemic stroke three days after AF ablation, presenting with left hemiparesis and dysarthria. Idarucizumab (5 g) enabled safe intravenous thrombolysis followed by emergency endovascular thrombectomy (EVT), achieving complete recanalization (mTICI 3). Angiography revealed an incidental 7 mm right MCA aneurysm at the occlusion site. Dabigatran was resumed on day 4, and one month later, the aneurysm was successfully treated with stent-assisted coil embolization. She remained asymptomatic at two months. **Conclusions:** This case illustrates how idarucizumab reversal expands reperfusion options by enabling both IVT and EVT in NOAC-treated patients and highlights the diagnostic role of EVT in revealing underlying vascular pathology, emphasizing the need for individualized post-procedural antithrombotic management.

## 1. Introduction

Atrial fibrillation (AF) is the most common sustained cardiac arrhythmia, with an increasing prevalence that poses a significant public health burden due to its association with a fivefold increased risk of stroke and a higher incidence of heart failure and all-cause mortality [1,2]. Catheter ablation has become a cornerstone in the management of AF, particularly for symptomatic patients refractory to antiarrhythmic drug therapy, and is a guideline-recommended first-line therapy for selected patients to improve symptoms and quality of life [3]. While AF ablation is a highly effective procedure, it is not without its risks. Embolic stroke, though rare, remains one of the most feared complications, with a reported incidence ranging from 0.1% to 2.3% especially in elderly populations in various studies [4,5]. The occurrence of a peri-procedural stroke is higher in the elderly, which can lead to devastating long-term neurological deficits, significantly impacting a patient’s life and increasing healthcare costs. The primary mechanism for stroke following AF ablation is believed to be thromboembolism, which can originate from the left atrium or pulmonary veins during or shortly after the procedure. To mitigate this risk, meticulous peri-procedural anticoagulation is essential. Historically, warfarin was the standard of care, but its use required careful monitoring of the international normalized ratio and often a bridging strategy with heparin, which increased the complexity and bleeding risk. The advent of non-vitamin K antagonist oral anticoagulants (NOACs) has revolutionized this management. Current guidelines and a growing body of evidence support the use of an uninterrupted NOAC strategy during the peri-procedural period of AF ablation, demonstrating similar or even lower rates of thromboembolic and major bleeding events compared to warfarin [6,7].

Acute management of ischemic stroke in patients on NOACs presents a unique challenge. Intravenous thrombolysis with recombinant tissue plasminogen activator (rt-PA) is contraindicated in the presence of therapeutic anticoagulation because of bleeding risk. Idarucizumab, a monoclonal antibody fragment, provides rapid and complete reversal of dabigatran, enabling timely administration of intravenous thrombolysis in otherwise ineligible patients [8]. In addition, mechanical endovascular thrombectomy (EVT) has become the standard of care for large-vessel occlusions, achieving high rates of recanalization and favorable outcomes when performed promptly [9].

Here, we report a rare case of embolic stroke occurring shortly after AF ablation despite uninterrupted dabigatran therapy. The patient achieved complete neurological recovery after dabigatran reversal, intravenous thrombolysis, and EVT. Notably, angiography revealed a previously unrecognized large middle cerebral artery (MCA) aneurysm, which was subsequently treated with stent-assisted embolization. This case highlights both the therapeutic value of rapid reversal and multimodal reperfusion strategies and the diagnostic role of EVT in uncovering clinically significant vascular pathology.

## 2. Case Report

A 49-year-old woman with long-standing atrial fibrillation, maintained on dabigatran 110 mg twice daily, underwent elective catheter ablation. The procedure included pulmonary vein isolation of all four veins and creation of a mitral isthmus line using radiofrequency energy delivered *via* a ThermoCool^®^ SmartTouch^®^ SF catheter (Biosense Webster, Inc., Irvine, CA, USA) with power settings of 45–50 W and a target ablation index (AI) of 450–500. Activated clotting time (ACT) was checked every 15 min, with only one value below 300 s (271 s). Pre-procedural transesophageal echocardiography confirmed no left atrial thrombus.

Pre-procedural transesophageal echocardiography confirmed the absence of left atrial thrombus. On the third post-procedural day, she developed sudden-onset left facial palsy, dysarthria, and limb clumsiness, with a National Institutes of Health Stroke Scale (NIHSS) score of 6. Head computed tomography (CT) excluded intracranial hemorrhage. Given her recent dabigatran use, she received an immediate intravenous bolus of idarucizumab (5 g) for anticoagulation reversal, which enabled subsequent administration of intravenous recombinant tissue plasminogen activator (rt-PA, 0.6 mg/kg).

On postoperative day 3, she developed sudden left facial palsy, dysarthria, and limb clumsiness (NIHSS 6). Head CT excluded hemorrhage. Because of recent dabigatran use, idarucizumab 5 g was administered for rapid anticoagulation reversal, enabling IV rt-PA (0.6 mg/kg). MRI/DWI revealed acute cortical and subcortical infarction in the right MCA territory (Figure 1A), and TOF-MRA confirmed right M1 occlusion (Figure 1B). Perfusion–diffusion analysis showed a core infarct of 22 mL and a hypoperfused region of 93 mL (mismatch ratio 4.2; salvageable penumbra 71 mL; Figure 2). Emergency EVT achieved complete recanalization (mTICI 3, [10]) (Figure 3A). (Figure 3A). Recanalization was achieved with full restoration of antegrade flow and MCA branch perfusion (modified Thrombolysis in Cerebral Infarction grade 3).

Post-recanalization angiography revealed a previously unrecognized 7.2 mm MCA bifurcation aneurysm with a 4.7 mm neck (Figure 3B). Three-dimensional rotational angiography confirmed a broad-neck bifurcation aneurysm (Figure 3C). During EVT, systolic BP was maintained at 120–140 mmHg to minimize rupture risk, with only minimal heparinization to maintain catheter patency. After confirming no hemorrhage on follow-up imaging, dabigatran was resumed on day 4. The patient was discharged on day 11 (NIHSS 1). One month later, she underwent stent-assisted coil embolization, achieving complete aneurysm obliteration with preserved MCA branch patency (Figure 3D). She remained asymptomatic with full neurological recovery at three-month follow-up. Retrieved thrombi were dark red with central pale foci, consistent with fibrin-rich cardioembolic composition (Figure 4).

## 3. Discussion

Catheter ablation has become an established treatment for AF, but peri-procedural thromboembolic events remain a feared complication even with uninterrupted NOAC therapy. Our case underscores that embolic stroke may still occur despite dabigatran use, highlighting the importance of vigilance in the early post-ablation period. Catheter ablation does not entirely eliminate the thromboembolic risk, thereby reinforcing the guideline recommendation for continued CHA_2_DS_2_-VASc stratification post-ablation [3,11]. The underlying mechanism for post-ablation embolic stroke is often multifactorial. In our patient, the retrieved thrombi during EVT were macroscopically consistent with a fibrin-rich, cardioembolic origin (dark red with central pale foci) (Figure 4) [12]. This aligns with prevailing theories that suggest atrial stunning, endothelial injury along ablation lines, and catheter-related thrombus formation are primary contributors. These findings suggest a procedural or cardiac source rather than large-artery atherosclerosis, consistent with prior literature on peri-procedural stroke in AF ablation [13,14]. In addition to atrial stunning and endothelial injury, the ablation modality itself may influence thromboembolic risk. Radiofrequency ablation (RFA) generates higher tissue temperatures and produces greater microbubble formation, which can lead to transient microvascular injury and microembolic phenomena. By contrast, cryoablation causes cellular necrosis through freezing-induced crystallization with minimal gas generation, thus potentially reducing embolic load. As reported by Anselmino et al., RFA has been associated with higher microbubble counts and a greater likelihood of subclinical cerebral emboli than cryoablation [13]. In our patient, the occurrence of ischemic stroke following RFA aligns with these mechanistic insights, suggesting that microbubble-mediated embolization could have contributed to the observed event.

This case highlights a major clinical advance—the safe use of multimodal reperfusion therapy in a patient anticoagulated with an NOAC. Although current guidelines generally contraindicate IVT in patients on therapeutic NOACs because of bleeding risk [15], the use of idarucizumab, a specific dabigatran reversal agent, enables rapid normalization of coagulation and allows safe thrombolysis [8,10,16]. The patient subsequently underwent EVT with complete recanalization (mTICI 3) and excellent neurological recovery, without hemorrhagic complications. Current recommendations support IVT followed by EVT in eligible cases [17], and endorse idarucizumab reversal prior to IVT [18]; however, its role before EVT remains less clearly defined. Our sequential approach—idarucizumab reversal, IVT, and EVT—illustrates a practical management pathway consistent with emerging case series and real-world experience [19]. According to the 2019 AHA/ASA guidelines, patients with large-vessel occlusion who receive intravenous thrombolysis should proceed to mechanical thrombectomy without delay once imaging confirms an accessible occlusion. Consistent with these recommendations and supported by Ghannam et al. (2023), immediate EVT following rt-PA (bridging therapy) achieves outcomes comparable to or better than direct EVT without higher hemorrhagic risk [18]. In our patient, rapid neurological deterioration and angiographic confirmation of an M1 occlusion justified prompt transition from IVT to EVT to secure early reperfusion. Compared with prior idarucizumab-assisted reperfusion reports—mostly involving spontaneous ischemic stroke under NOAC therapy [16,19]—our case occurred immediately after atrial-fibrillation ablation and was further complicated by an incidentally detected moderate-sized MCA aneurysm. This combination adds diagnostic and procedural complexity not previously described, thereby extending current understanding of idarucizumab-facilitated multimodal reperfusion in post-ablation stroke with unrecognized intracranial aneurysm.

Peri-procedural hemodynamic and antithrombotic management is crucial during EVT to minimize hemorrhagic complications. The 2019 AHA/ASA stroke guideline recommends maintaining BP ≤ 180/105 mmHg during and after reperfusion therapy [15]. Recent evidence suggests that stricter control—systolic BP 120–140 mmHg—may improve outcomes and reduce hemorrhage risk, particularly in patients with fragile vasculature or coexisting aneurysms [20]. Accordingly, systolic BP was maintained between 120 and 140 mmHg peri-procedurally in our patient to balance rupture prevention with adequate cerebral perfusion.

Antithrombotic therapy was individualized. Minimal heparinization was used intra-procedurally, with no additional antiplatelet loading. Dabigatran was resumed after imaging confirmed the absence of hemorrhage, followed by stepwise de-escalation based on thrombotic and bleeding risk. This tailored approach aligns with current expert recommendations for neurointerventional antithrombotic management [21].

A notable feature of this case was the incidental discovery of a 7 mm MCA aneurysm at the occlusion site during EVT, underscoring the dual diagnostic and therapeutic role of thrombectomy. According to standard size classifications (<10 mm = small/moderate; ≥10 mm = large) [22], unruptured aneurysms ≤10 mm generally do not increase hemorrhagic risk during IVT or EVT [23]. Consistent with this, our patient experienced no hemorrhagic complications. Current AHA/ASA guidelines state that small or asymptomatic aneurysms are not a contraindication to IVT, and both IVT and EVT would have been appropriate even if the lesion had been known beforehand [24]. The aneurysm was later treated with elective stent-assisted embolization, securing the lesion and reducing future rupture risk. This case further demonstrates that EVT can reveal unrecognized vascular pathology and guide long-term management [25].

Evidence from major PCI and CAS trials informs current neurointerventional antithrombotic strategies. The RE-DUAL PCI and AUGUSTUS studies showed that NOAC-based dual therapy significantly reduces major bleeding compared with triple therapy while maintaining similar ischemic protection [26,27,28]. Using Taiwan’s NHIRD, Huang et al. reported that oral anticoagulation alone yielded the lowest mortality after carotid stenting, whereas combination therapy increased bleeding without clear additional benefit [29]. Although extrapolated from coronary and carotid data, these findings support a pragmatic balance between thromboembolic protection and hemorrhagic risk in neurointerventional practice. In our patient, a three-month triple regimen with dabigatran, aspirin, and clopidogrel was well tolerated without bleeding, suggesting that short-term, individualized triple therapy may be appropriate in select high-risk cases.

In summary, this case highlights three key points: (1) embolic stroke can occur soon after AF ablation despite NOAC use, and (2) idarucizumab reversal enables safe multimodal reperfusion therapy with excellent outcomes; and (3) EVT not only restores perfusion but may also reveal clinically relevant vascular pathology requiring complex follow-up management. These findings underscore the importance of individualized, multidisciplinary care for this high-risk population.

## 4. Conclusions

This case demonstrates that embolic stroke may occur after AF ablation despite uninterrupted NOAC therapy. Rapid dabigatran reversal enabled safe intravenous thrombolysis followed by successful EVT, resulting in excellent neurological recovery. Notably, EVT not only restored cerebral perfusion but also incidentally revealed an unruptured MCA aneurysm, which was subsequently secured with stent-assisted embolization. The patient tolerated three months of triple therapy with dabigatran, aspirin, and clopidogrel without complications. These observations highlight the therapeutic value of rapid NOAC reversal, the dual diagnostic and therapeutic role of EVT, and the importance of tailored antithrombotic management, reinforcing the need for multidisciplinary strategies in complex stroke patients.

## Figures and Tables

**Figure 1 neurolint-17-00190-f001:**
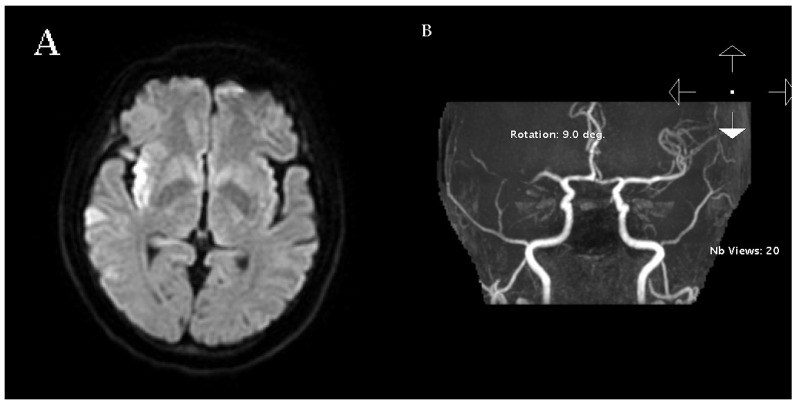
(**A**) Axial diffusion-weighted MRI showing cortical and subcortical hyperintensity in the right MCA territory, consistent with acute infarction. (**B**) Time-of-Flight Magnetic Resonance Angiography (TOF-MRA) showing occlusion of the right middle cerebral artery.

**Figure 2 neurolint-17-00190-f002:**
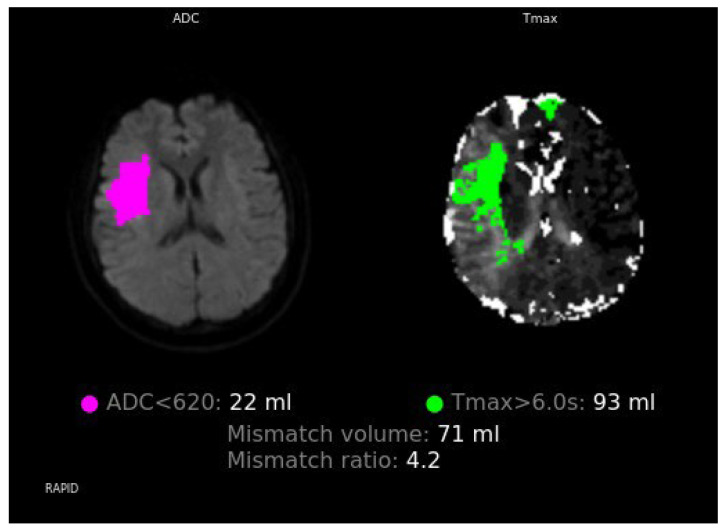
RAPID MRI demonstrating a perfusion–diffusion mismatch, with a core infarct volume of 22 mL (apparent diffusion coefficient < 620 × 10^−6^ mm^2^/s) and a hypoperfused region of 93 mL (Tmax > 6.0 s), yielding a mismatch ratio of 4.2 and a salvageable penumbra of 71 mL.

**Figure 3 neurolint-17-00190-f003:**
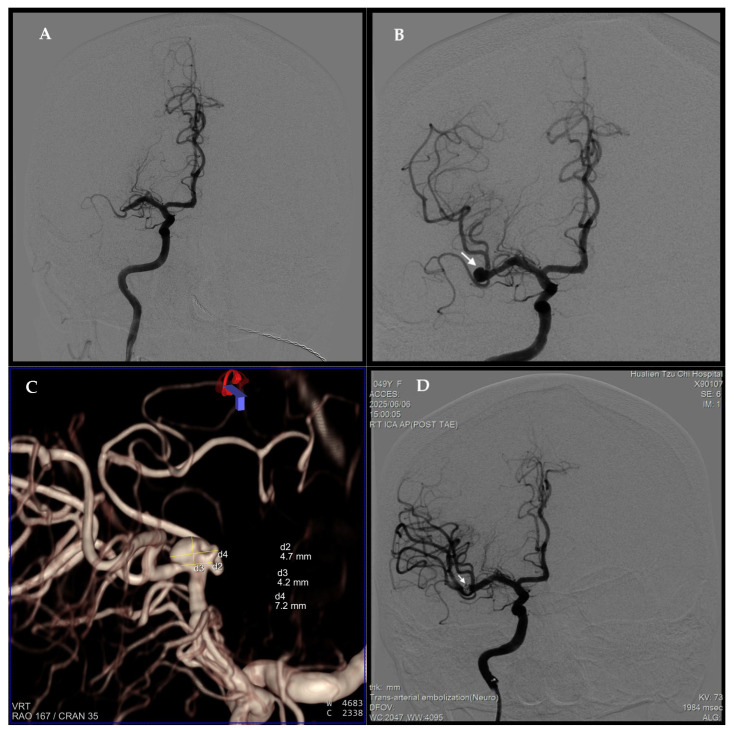
(**A**) Pre-thrombectomy angiogram showing distal ICA/proximal MCA occlusion. (**B**) Post-thrombectomy angiogram demonstrating flow restoration (mTICI 3) and incidental visualization of a distal MCA bifurcation aneurysm (arrow). (**C**) 3D rotational angiography confirming a broad-neck aneurysm (7.2 mm maximal diameter, 4.7 mm neck). (**D**) Angiogram after stent-assisted coil embolization showing complete aneurysm occlusion with preserved vessel patency (arrow).

**Figure 4 neurolint-17-00190-f004:**
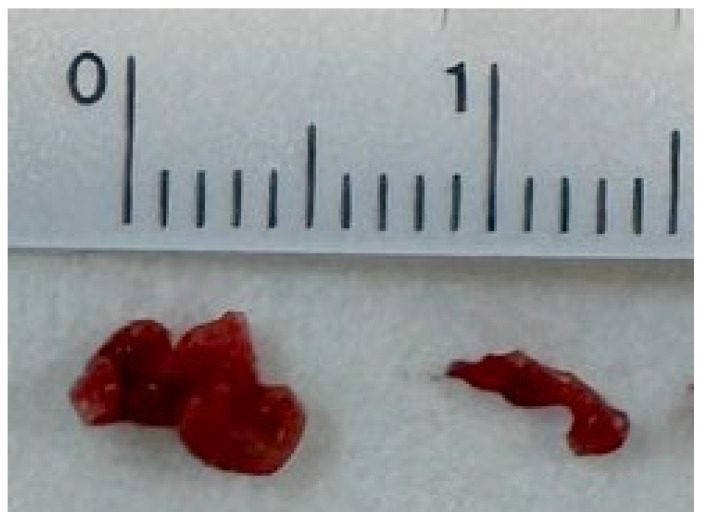
Two larger thrombotic fragments obtained during thrombectomy, showing central pale spots consistent with fibrin-rich cores, supporting a cardioembolic source.

## Data Availability

The clinical data and imaging materials used in this case report are not publicly available due to patient privacy and confidentiality restrictions. However, anonymized data may be made available from the corresponding author upon reasonable request and with approval from the relevant institutional ethics committee.

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
