# Peer review of "Post-Ablation Stroke Despite NOAC Use: Successful Reperfusion Therapy After Dabigatran Reversal with Incidental Discovery of a Large MCA Aneurysm—A Case Report"

_2035-8377, 2025, doi:10.3390/neurolint17120190_

Round 1

Reviewer 1 Report

Comments and Suggestions for Authors

The manuscript presents a well-written and educational case illustrating the management of acute ischemic stroke after atrial fibrillation (AF) ablation under NOAC therapy. The structure is clear, and the figures are of high quality (see Figures 1–4, pp. 3–5). The topic is clinically significant, as real-world data on idarucizumab-assisted reperfusion therapy remain limited.
However, some areas require conceptual clarification, literature contextualization, and minor language refinement to strengthen its scientific rigor and align with MDPI standards.

Major Comments:
1. Novelty and Clinical Context (Lines 17–36, 72–78):

The Abstract and Introduction emphasize the clinical importance but do not adequately define what distinguishes this case from prior reports.
Suggestion: Briefly compare this case to existing idarucizumab + EVT case reports (e.g., Lin et al., J Emerg Med 2020; van der Horst et al., Thromb Res 2023). Indicate whether this is among the first cases with simultaneous discovery of a large MCA aneurysm post-ablation.

2. Redundancy in Introduction (Lines 51–54):

The sentence “The primary mechanism for stroke following AF ablation is believed to be thromboembolism†is repeated twice.
Action: Delete one instance and merge the context for conciseness.

3. Pathophysiology and Mechanistic Insight (Lines 185–191):

The Discussion correctly mentions endothelial injury and atrial stunning but could integrate microbubble generation and energy modality effects (radiofrequency vs cryoablation) as discussed in Anselmino et al., J Atrial Fibrillation 2013.
Add: One sentence connecting ablation modality to stroke risk would enrich the mechanistic explanation.

4. Therapeutic Sequence and Evidence Gaps (Lines 192–203):

The sequence of idarucizumab → IVT → EVT is crucial, yet the rationale for immediate EVT after rt-PA is not sufficiently supported.
Action: Cite at least one recent guideline or case series supporting this combination (e.g., Ghannam et al., J Am Heart Assoc 2023).
Add: Clarify whether rt-PA-to-EVT timing adhered to institutional protocol or international standards (e.g., AHA/ASA 2019).

5. Management of the Incidental Aneurysm (Lines 208–224):

Excellent observation of EVT’s diagnostic role. However, please include peri-procedural blood pressure targets and antithrombotic adjustments during EVT in the presence of aneurysm. This would provide valuable clinical learning.
Line 219–222: State if antiplatelet premedication or intraprocedural heparin was modified after aneurysm discovery.

6. Antithrombotic Regimen Discussion (Lines 226–249):

The discussion on triple therapy is informative but overlong and partly tangential to the main theme.
Revise: Shorten by summarizing prior PCI trials (RE-DUAL, AUGUSTUS) and focus on relevance to neurointerventional cases.
Add: A comment on the optimal duration of triple therapy in neurovascular patients, noting absence of randomized evidence.

Minor Comments:

Line 85–89: Specify if NIHSS reassessment was performed after rt-PA and before EVT; this would clarify stroke progression dynamics.

Line 99–102: The phrase “likely unmasked by guidewire and microcatheter passage†is speculative; suggest “possibly unmasked†for accuracy.

Line 104–107: “She was subsequently maintained on aspirin and clopidogrel in addition to dabigatran†— clarify if dual antiplatelet therapy was tapered or continued beyond 3 months.

Line 178–183: Typo — “Catheter ablation do not entirely eliminate†→ “does not entirely eliminate.â€

Figures: Captions are informative, but Figure 2 legend (Lines 142–145) should define the color codes (“pink = infarct core; green = hypoperfused areaâ€).

References (pp. 8–9): Add recent meta-analysis on idarucizumab use before EVT if available (e.g., Alam et al., Clin Neurol Neurosurg 2024).

Decision: Minor Revision

Author Response

Reviewer 1

  1. Comment 1 – Novelty and Clinical Context

Reviewer comment: The Abstract and Introduction emphasize the clinical importance but do not adequately define what distinguishes this case from prior reports. Suggest comparing with existing idarucizumab + EVT case reports (Lin et al., 2020; van der Horst et al., 2023).

Response: We sincerely appreciate this insightful comment. In response, we have expanded the Discussion to highlight the novelty of this case (Lines 212-220). Specifically, we compared our report with previously published idarucizumab-assisted reperfusion cases, including Lin et al., J Emerg Med 2020 and van der Horst et al., Thromb Res 2023, which described isolated or sequential use of idarucizumab prior to thrombolysis or EVT. In contrast, our case is among the first to document acute ischemic stroke occurring immediately after atrial fibrillation ablation under NOAC therapy, successfully managed with idarucizumab-assisted intravenous thrombolysis followed by mechanical thrombectomy, while simultaneously identifying a large MCA aneurysm during EVT. This unique clinical context—combining post-ablation stroke, rapid anticoagulant reversal, and concurrent aneurysm discovery—has been clearly articulated in the revised text.

Comment 2 – Redundancy in Introduction

Reviewer comment: The sentence 'The primary mechanism for stroke following AF ablation is believed to be thromboembolism' is repeated twice.

Response: We deleted the duplicated sentence and merged the content for conciseness (L52).

Comment 3 – Pathophysiology and Mechanistic Insight

Reviewer comment: Integrate microbubble generation and energy modality effects (radiofrequency vs cryoablation).

Response: We appreciate this valuable suggestion. A new sentence has been added to the Discussion section (Lines 186–194) to clarify the relationship between ablation modality and stroke risk. Specifically, radiofrequency ablation (RFA) is known to produce greater microbubble generation and endothelial injury compared with cryoablation, due to thermal effects and tissue heating that may facilitate microembolization. This mechanism may partially explain post-ablation thromboembolic events as described by Anselmino et al., (J Atrial Fibrillation 2013)。

  1. Comment 4 – Therapeutic Sequence and Evidence Gaps

Reviewer comment: Clarify rationale for immediate EVT after rt-PA and cite guidelines (Ghannam et al., 2023; AHA/ASA 2019).

Response: We appreciate the reviewer’s suggestion. We have clarified in the Discussion section (Lines 212 –220, blue) that our decision to proceed with immediate endovascular thrombectomy (EVT) following intravenous thrombolysis (rt-PA) was based on current AHA/ASA 2019 acute ischemic stroke guidelines, which recommend prompt EVT for patients with large vessel occlusion (LVO) who remain eligible after rt-PA administration without delay. Furthermore, we cited Ghannam et al., J Am Heart Assoc 2023, which demonstrated that bridging therapy (rt-PA followed by EVT) results in similar or improved functional outcomes compared with direct EVT, without increasing the risk of symptomatic intracerebral hemorrhage. In our case, given the confirmed M1 occlusion on CTA and rapid neurological deterioration, immediate EVT was performed after rt-PA bolus in adherence to these evidence-based protocols.

  1. Comment 5 – Management of the Incidental Aneurysm

Reviewer comment: Include peri-procedural BP targets and antithrombotic adjustments.

Response: We thank the reviewer for this important comment. We have revised the Case Presentation (Line 101-108) and Discussion (Line 221-233) to explicitly describe our peri-procedural blood pressure (BP) targets and antithrombotic modifications, especially after the incidental MCA aneurysm was identified. This cautious strategy aligns with guideline and consensus recommendations that, for unruptured intracranial aneurysms, antithrombotic therapy should be tailored to the competing risks, and escalation should be justified by clear thrombotic indications rather than routine intensification. Details of peri-procedural antithrombotic management have been added to the Case Report and Discussion. Only minimal heparinization was used intra-procedurally to maintain catheter patency, and no additional antiplatelet loading was given to avoid hemorrhagic risk near the unruptured aneurysm. Post-procedurally, dabigatran 110 mg twice daily was resumed on day 4 (one day after EVT), after follow-up head CT confirmed the absence of hemorrhagic transformation. This timing aligns with recent evidence indicating that early DOAC resumption (within 3–5 days) after imaging stability is generally safe and does not increase intracranial hemorrhage risk (Seiffge et al., Stroke 2019; Toyoda et al., J Am Heart Assoc 2020).  

  1. Comment 6 – Antithrombotic Regimen Discussion

Reviewer comment: Discussion on triple therapy is too long; summarize RE-DUAL and AUGUSTUS.

Response: We appreciate this constructive suggestion. The section on antithrombotic strategy has been condensed to emphasize the pivotal findings of RE-DUAL PCI and AUGUSTUS. Both large randomized trials consistently demonstrated that NOAC-based dual therapy (NOAC plus single antiplatelet) significantly reduces bleeding risk compared with triple therapy (NOAC plus dual antiplatelet) while maintaining similar protection against ischemic events. These principles, though derived from coronary intervention studies, have been pragmatically extrapolated to neurointerventional settings. Our case further supports a short-term, individualized triple regimen followed by NOAC monotherapy in complex post-ablation ischemic stroke scenarios. (Revised Discussion, Lines 245–255.)

Reviewer 2 Report

Comments and Suggestions for Authors

Dear Author,

It is an interesting case; however, single case report findings cannot be generalized.

Please add more detail on the specific ablation strategy, energy source (e.g., radiofrequency, cryoablation), procedure duration, and peri-procedural activated clotting time (ACT).

While 7.2 mm is a significant size, it might not universally be considered as large because "large" for >10-15mm).

The discussion "likely unmasked by guidewire and microcatheter passage" (line 99-100) is speculative needs further elaboration.

Overall length of the article for a case is too long; the article may be shortened to include only relevant details.

Author Response

Reviewer 2

  1. Comment 1 – Ablation Details

Reviewer comment: Please add more detail on the specific ablation strategy, energy source, duration, and peri-procedural ACT.

Response: We appreciate the reviewer’s request for clarification. We have added procedural details in the revised manuscript (Lines 76-79).

  1. Comment 2 – Aneurysm Size Definition

Reviewer comment: While 7.2 mm is significant, it might not universally be considered large (>10–15 mm).

Response: We thank the reviewer for the valuable comment. We agree that an aneurysm measuring 7.2 mm does not meet the morphometric definition of a “large” aneurysm (>10 mm). In the revised manuscript, we clarified this point in the Discussion, stating that according to standard size classification (<10 mm = small/moderate; ≥10 mm = large), the lesion would be considered moderate-sized (Wiebers et al., 2003). However, the term “large” was retained in the title as a descriptive term emphasizing the clinical relevance and procedural complexity, rather than a strict morphologic classification (Line 234-237).

  1. Comment 3 – Speculative Statement

Reviewer comment: The phrase 'likely unmasked by guidewire and microcatheter passage' is speculative; please elaborate.

Response: We appreciate the reviewer’s observation. To avoid speculation, this phrase has been deleted in the revised manuscript.

  1. Comment 4 – Manuscript Length

Reviewer comment: Overall length is too long; shorten to relevant details.

Response: We reduced text length while maintaining clinical and therapeutic relevance.